# Potential of Trilayered Gelatin/Polycaprolactone Nanofibers for Periodontal Regeneration: An In Vitro Study

**DOI:** 10.3390/ijms26020672

**Published:** 2025-01-15

**Authors:** Zhiwei Tian, Zhongqi Zhao, Marco Aoqi Rausch, Christian Behm, Dino Tur, Hassan Ali Shokoohi-Tabrizi, Oleh Andrukhov, Xiaohui Rausch-Fan

**Affiliations:** 1Competence Center for Periodontal Research, University Clinic of Dentistry, Medical University of Vienna, 1090 Vienna, Austria; n11945430@students.meduniwien.ac.at (Z.T.); marco.rausch@meduniwien.ac.at (M.A.R.); christian.behm@meduniwien.ac.at (C.B.); 2Clinical Division of Orthodontics, University Clinic of Dentistry, Medical University of Vienna, 1090 Vienna, Austria; 3Division of Conservative Dentistry and Periodontology, University Clinic of Dentistry, Medical University of Vienna, 1090 Vienna, Austria; dino.tur@meduniwien.ac.at (D.T.); xiaohui.rausch-fan@meduniwien.ac.at (X.R.-F.); 4Core Facility Applied Physics, Laser and CAD/CAM Technology, University Clinic of Dentistry, Medical University of Vienna, 1090 Vienna, Austria; hassan.shokoohi-tabrizi@meduniwien.ac.at; 5Center for Clinical Research, University Clinic of Dentistry, Medical University of Vienna, 1090 Vienna, Austria

**Keywords:** periodontal regeneration, nanofiber, polycaprolactone, collagen, human periodontal ligament-derived stromal cells

## Abstract

Over the past few years, biomaterial-based periodontal tissue engineering has gained popularity. An ideal biomaterial for treating periodontal defects is expected to stimulate periodontal-derived cells, allowing them to contribute most efficiently to tissue reconstruction. The present study focuses on evaluating the in vitro behavior of human periodontal ligament-derived stromal cells (hPDL-MSCs) when cultured on gelatin/Polycaprolactone prototype (GPP) and volume-stable collagen matrix (VSCM). Cells were cultured onto the GPP, VSCM, or tissue culture plate (TCP) for 3, 7, and 14 days. Cell morphology, adhesion, proliferation/viability, the gene expression of Collagen type I, alpha1 (COL1A1), Vascular endothelial growth factor A (VEGF-A), Periostin (POSTN), Cementum protein 1 (CEMP1), Cementum attachment protein (CAP), Interleukin 8 (IL-8) and Osteocalcin (OCN), and the levels of VEGF-A and IL-8 proteins were investigated. hPDL-MSCs attached to both biomaterials exhibited a different morphology compared to TCP. GPP exhibited stronger capabilities in enhancing cell viability and metabolic activity compared to VSCM. In most cases, the expression of all investigated genes, except POSTN, was stimulated by both materials, with GPP having a superior effect on COL1A1 and VEGF-A, and VSCM on OCN. The IL-8 protein production was slightly higher in cells grown on VSCM. GPP also exhibited the ability to absorb VEGF-A protein. The gene expression of POSTN was promoted by GPP and slightly suppressed by VSCM. In summary, our findings indicate that GPP electrospun nanofibers effectively promote the functional performance of PDLSCs in periodontal regeneration, particularly in the periodontal ligament and cementum compartment.

## 1. Introduction

Defects of periodontal tissue are mostly caused by periodontitis, one of the most prevalent inflammatory diseases in the population [1,2]. Periodontal surgical approaches have considerable therapeutic advantages and typically require creating surgical access to the root surface, enabling the comprehensive removal of plaque and calculus on the one hand, and providing conditions for the implantation of various periodontal regeneration materials on the other hand [3]. This treatment approach results in not merely periodontal tissue repair; it also enables the regeneration of periodontium and reconstruction of their original structure and function [4,5].

For decades, a wide range of treatment methods has been utilized in efforts to improve reattachment in periodontal tissues, with varying degrees of potency [6,7], but still, only incomplete regeneration of a damaged periodontium is currently achievable in clinical practice [8]. Periodontal regeneration requires timely and spatially coordinated healing processes and involves the regeneration of various hard and soft tissues. Periodontal regeneration therapy has incorporated numerous biological materials, which have had a beneficial influence on tissue formation. Presently, a prevalent strategy involves the utilization of diverse biocompatible membranes to establish a conducive environment between the root surface and the materials, facilitating the desired tissue formation and consequently achieving successfully guided tissue regeneration [9].

The application of three-dimensional scaffolds is an emerging trend in periodontal regeneration with promising perspectives [10,11]. These scaffolds serve as mechanical support, affect the resident cells, and promote their differentiation into various types. Modifying scaffold properties, like stiffness, chemical composition, and porosity, can modulate these processes [12,13]. Some scaffolds, particularly volume-stable collagen matrix (VSCM), stimulate soft-tissue remodeling and angiogenesis and demonstrate a promising potential for periodontal regeneration [14,15]. Nevertheless, collagen-based materials have limitations, including relatively high costs, intricate manufacturing processes, limited viscosity, and potential constraints stemming from religious and cultural considerations [16,17]. Gelatin (GE), a product of collagen hydrolysis, retains its biocompatibility while reducing immunogenicity and is considered a favored resource in the medical field. Due to a lower molecular weight, GE is easier to manipulate using methods like electrospinning. Still, GE also has the drawback of inadequate mechanical properties [18,19].

The mechanical properties of GE might be improved by combining it with synthetic polymers, such as polycaprolactone (PCL), which not only possess outstanding mechanical properties and an extremely high elastic modulus but also offer commendable biocompatibility [20]. Recently, a prototype combining GE and PCL (GPP) showcased suitable biocompatibility, elasticity, and capability of maintaining structural integrity in an aqueous environment, significantly enhancing wound healing in an animal study, making it a promising material for tissue regeneration [21,22,23]. Previous studies in vitro demonstrated that GPP electrospun nanofibers positively affect the regenerative potential of gingival cells and have great potential for periodontal soft-tissue regeneration [21,24]. However, the potential of this material to regenerate complex periodontal structures has yet to be investigated.

In the present study, we focused on the effect of the GPP prototype on human periodontal ligament mesenchymal stromal cells (hPDL-MSCs). These cells have the potential to regenerate complex periodontal structures, particularly cementum-like tissues, bone, and collagen fibers akin to Sharpey’s fibers, thus restoring functional periodontal attachment [25]. Particularly, this study focused on in vitro investigating the effects of GPP on the attachment, proliferation, and metabolic activity of hPDL-MSCs, as well as the production of biomolecules and proteins pertinent to the periodontal regeneration process. The effect of GPP was compared to that of VSCM.

## 2. Results

### 2.1. Cell Morphology and Attachment

The cell growth patterns, and the specific structural features of the scaffolds, are depicted in SEM Figure 1. After 3 days of culture, hPDL-MSCs on GPP exhibited efficient proliferation and spread, predominantly assuming in spindle and polygonal shapes (Figure 1a). Numerous adhesion sites between cells and nanoscaled fibers were revealed under high-magnification images, with cells penetrating the material structure and integrating beneath the superficial fiber layer (Figure 1b). After 1 week, the surface of GPP was mostly covered by hPDL-MSCs. Observation at 1500x magnification demonstrated an interaction of hPDL-MSCs with the fibrous elements of GPP (Figure 1c,d). By the end of the 2-week culture period, GPP was covered by a cohesive and dense (multi)layer of hPDL-MSCs (Figure 1e,f). It was difficult to distinguish cells attached to VSCM on days 3 and 7, probably due to their migration inside the pores and their discrimination from the scaffold structures (Figure 1g–j). However, a continuous layer of interacting cells was established after 2 weeks, overlaying several of the microporous areas of the scaffold (Figure 1k,l).

The visualization of actin filament and cell nuclei of hPDL-MSCs grown on various materials is shown in Figure 2. Cells grown on TCP exhibited a typical fibroblast-like morphology after 3 days of culture (Figure 2c). In contrast, hPDL-MSCs on GPP had both spindle- and polygonal-like morphologies and demonstrated an appearance of some branching structures (Figure 2a). Cells on TCP were almost confluent at this time point. In comparison to the other two substrates, VSCM was relatively scarcely populated by hPDL-MSCs, appearing in either spindle or polygonal shapes (Figure 2b). Upon 7 days of cell seeding, a marked enhancement in the cell density was noted among all three groups. Cells cultured on GPP and TCP approached full coverage of the material surfaces (Figure 2d,f). The observation of cells on VSCM was complicated by stained cells, which migrated within the membrane and appeared as cloud-like structures (Figure 2e). After 14 days of culture, GPP, and TCP were fully covered by hPDL-MSCs (Figure 2g,i), whereas the coverage of VSCM by the cells was still incomplete (Figure 2h). Moreover, compared to TCP, the cytoskeleton arrangement on GPP tended to be more paralleled oriented. At this stage, cells on VSCM presented a tendency to bridge and proliferate across the microporous structures of the scaffold. Compared to their morphology at 3 and 7 days, the cells further flattened and stretched; the intercellular connections intensified, forming a network structure without directional preference in cell alignment.

### 2.2. Cell Viability

The metabolic activity of hPDL-MSCs grown on different materials and measured using the CCK-8 and resazurin assays are presented in Figure 3 and Figure 4. Over the whole observation period, the metabolic activity of hPDL-MSCs grown in different conditions gradually increased, suggesting continuous cell proliferation and cells on TCP exhibited a significantly higher metabolic activity than those grown on two other substrates (*p* < 0.05). Additionally, the metabolic activity of hPDL-MSCs grown on GPP on cell proliferation was significantly higher than VSCM at all time points except CCK-8 assay after 14 days (*p* < 0.05). After 14 days of cell seeding, although the detection level of CCK-8 from nanofiber and VSCM no longer showed significant differences, cells grown on the former demonstrated a faster proliferation trend (*p* = 0.056).

### 2.3. Effect of GPP and VSCM on Gene Expression Level of Various Biomarkers

The influence of different substrates on the gene expression levels of various biomarkers is presented in Figure 5. The expression of COL1A1, CEMP-1, CAP, and VEGF-A was markedly enhanced in cells grown on both scaffolds compared to TCP (Figure 5a,c,d,e), in most cases, statistically significant differences were observed. Moreover, hPDL-MSCs grown on GPP exhibited higher gene expression levels of COL1A1 (Figure 5a, *p* < 0.05 at days 3 and 14) and VEGF-A (Figure 5e, *p* < 0.05 at days 3 and 7) compared to those grown on VSCM. GPP notably enhanced periostin (POSTN) expression levels in hPDL-MSCs over VSCM after 7 and 14 days (Figure 5b, *p* < 0.05), whereas VSCM decreased POSTN expression after day 7 (Figure 5b, *p* < 0.05). The gene expression of inflammatory protein IL-8 was generally higher in cells grown on the membranes compared to TCP, but no differences between GPP and VSCM were observed (Figure 5f). The gene expression of osteogenesis-related protein OCN was not detectable in many samples after 3 and 7 days. After 14 days, the expression levels of OCN were significantly higher in cells on VSCM compared to that of cells cultured on GPP and TCP (Figure 5g, *p* < 0.05).

### 2.4. Effects of GPP and VSCM on the Production of Related Protein

The amount of IL-8 and VEGF-A production in conditioned media is displayed in Figure 6 and Figure 7a. Within 14 days of culturing, hPDL-MSCs cultured on GPP and VSCM generated significantly more VEGF-A protein in contrast to those on TCP (Figure 7a, *p* < 0.05). Furthermore, the amount of VEGF-A in cells grown on VSCM was significantly higher than in GPP throughout the observation period. In terms of IL-8, although two biomaterials seemed to stimulate the synthesis of this chemokine, notable differences were only significant between cells cultured on VSCM and those on TCP at 3 and 14 days.

### 2.5. VEGF Protein Absorption

Figure 7b shows the content of recombinant VEGF165 protein after 6 h incubation with and without materials. The content of VEGF in the conditioned media was significantly lower in wells without any material compared to the wells, in which one of the materials was present (*p* < 0.01). Furthermore, the content of VEGF in wells containing VSCM was significantly higher than in those containing GPP (*p* < 0.05). The amount of VEGF released within 24 h from the materials soaked with 10 ng/mL of recombinant VEGF 165 protein is presented in Figure 7c. The amount of VEGF released from VSCM was significantly higher than that from GPP (*p* < 0.01).

## 3. Discussion

Given the inherent shortcomings of autologous materials, xenogeneic natural materials have been considered as alternatives. Exploring different natural materials within the tissue engineering field primarily seeks to bio-mimic the natural ECM environment more accurately and intends to facilitate genuine tissue regeneration [26]. In addition, the limited sources of hPDL-MSCs play a primary role in the regeneration of periodontium, particularly periodontal ligament, and therefore, understanding their interaction with various materials is crucial to estimating their potential clinical effectiveness. In this study, we investigated the attachment, growth, and expression of functional parameters in hPDL-MSCs grown on two xenogeneic-derived scaffolds developed for application in periodontal regenerative medicine. The healing process after periodontal surgery, like any wound healing, consists of four overlapping phases: hemostasis, inflammation, proliferation, and tissue remodeling. On the cellular level, the following events occur: cell migration to the surgery area and attachment to the scaffold, their proliferation, production of various biologically active proteins, and differentiation to the functional cells.

The first prerequisite for any three-dimensional (3D) scaffold is an effective cell attachment and establishing a cellular network of interacting cells [27]. SEM and fluorescence microscopy observations revealed the continued growth of hPDL-MSCs on both scaffolds. Regarding the expansion area, cells on GPP performed better than those on VSCM. After 3 days, SEM showed that cells grown on GPP were tightly intertwined with surrounding nanofiber structures. SEM observation from day 14 indicated that hPDL-MSCs cultured on two types of materials encased by continuous sheet-like structures which tended to cover the material’s surfaces, implying the possible formation of ECM-like structures. The presence of nanofibers in GPP seems to be an essential factor influencing hPDL-MSCs attachment, as they provide the attachment points and also influence cell morphology and orientation. Similarly, previous researchers discovered that the arrangement of PDL stem cells was determined by the orientation of PCL/GE fibers [28]. In contrast, visual assessments indicated fewer cells on VSCM at the early stage of culture. This can be explained by the pore size of VSCM, which was comparable with the size of single cells and thus allowed potential cell migration within a scaffold. Thus, the scaffold architecture and structural features are essential for the initial interaction with cells: nanofibers facilitate cell attachment and might be used for cell spatial orientation, whereas the presence of microscale pores allows cell migration inside the scaffold. After a 2-week prolonged culturing, both GPP and VSCM scaffolds were covered by cells, but some differences in the coverage degree and cell morphology were still observed.

The simple presence of cells on materials does not reflect their metabolic integrity, and cell proliferation and metabolic activity serve as fundamental metrics for evaluating the biological performance of scaffolds. Over a 2-week culture period, hPDL-MSCs grew consistently on both scaffolds. However, the growth was less efficient compared to tissue culture plastic. Cells on GPP exhibited notably higher proliferation and metabolic activity than those on VSCM, especially in the first week. This outcome is in line with the recent study showing that PDL stem cells achieved high proliferation rates on hybrid PCL/GE materials [28]. It is mentionable that the nanoscale surface structure of GE-blended scaffolds could significantly enhance the proliferation of PDL stem cells [29]. Cell proliferation/viability and metabolic activity results correspond with the findings of microscopy experiments, showing a higher cell density on GPP compared to VSCM.

Periodontium is a complex heterogeneous structure consisting of a combination of hard and soft tissues. An effective periodontal regeneration requires the regeneration of all types of tissues, which makes fabricating the appropriate scaffold quite challenging. Therefore, we have investigated the effect of two scaffolds on the gene expression of various factors associated with the formation of different periodontal tissues. Periostin is specifically present in the periodontal ligament, where it fulfills a variety of functions in tissue development and periodontal disease [30]. We found that GPP had a more favorable effect on periostin expression in hPDL-MSCs than cells on VSCM after 7 days of culture, which implies that GPP has a stronger influence in prompting the hPDL-MSCs to differentiate into periodontal ligament cell populations. This finding agrees with recent research demonstrating that the trilayered bioscaffold produced from hybrid PCL and GE increased the expression of POSTN in PDL stem cells after 14 days of culture [28].

In addition to the ligament region, restoration and reconstruction of the cementum tissue are essential for ensuring the integrity of attachment structures. The innate capacity of cementum repair is limited, and when the structures are absent and replaced by fibrous or bone tissue, root resorption or ankylosis is likely to occur [31]. CAP facilitates the attachment of collagen to the root surface and influences the migration and adherence of surrounding ligament fibroblasts [32,33]. CEMP1 is known to induce the differentiation of multiple cell types into cementoblast-like phenotypes and promote the formation of new cementum [34]. In this study, both biomaterials effectively promoted the expression of CEMP1 and CAP, from the early stage of culture, suggesting that they have the potential to support cementoblasts-like cell differentiation and cementogenesis. The earlier study has also shown that electrospun scaffolds combining GE and PCL enhance the expression levels of these proteins in PDL stem cells by day 7 [28].

The alveolar bone, being the principal supportive tissue of the periodontal complex, is regarded as the indispensable basis for both the functional and structural aspects of periodontal regeneration. Osteocalcin, as a key factor in the late stage of osteogenesis, is closely linked to the mineralization of bone tissue and the homeostasis of calcium ions [35]. Alvarez Perez et al. found that hMSCs grown on PCL/GE electrospun scaffold revealed increased OCN expression after 1 week [36]. We did not detect the OCN expression within the first week, which can be because hPDL-MSCs are progenitor undifferentiated cells. Moreover, a previous study showed that CEMP1 might downregulated OCN expression in PDL cells [31]. However, by day 14, both biomaterials seemed to have a positive regulatory effect on OCN expression, with VSCM showing a more noticeable impact. This observation might imply that at this point, hPDL-MSCs start to differentiate into bone cells, with VSCM showing a more noticeable impact than GPP.

Collagen 1 is the primary organic component of ligaments, bone, and cementum; the proportion of COL1 in periodontal ligament even determines the health and functionality of the ligament’s structure [31,37]. In our research, GPP displayed a notable stimulating effect on COL1A1 expression in hPDL-MSCs over the 2-week culture time, significantly higher than VSCM’s. Correspondingly, the previous study on PCL mixed with GE has reported an increase in COL1A1 expression in hMSCs within the first week of culture [36].

Angiogenesis is an essential prerequisite for periodontal regeneration because it is crucial for delivering oxygen and other biomolecules within the tissue [38]. As a crucial paracrine factor secreted by MSCs, VEGF-A effectively promotes angiogenesis and plays a regulatory role in bone formation [39]. In the current study, the expression levels of the VEGF-A gene were elevated in both material groups, with a more pronounced enhancement in the GPP group during the first week. This aligns with a prior study involving Poly (D, L-lactide-co-glycolide)/GE electrospun scaffolds, where the materials stimulated VEGF-A expression in dental follicle stem cells after 7 days [40]. Likewise, the histological observations have indicated that PCL/GE nanofibers contribute to new blood vessel formation at the periodontal defect site [41].

Intriguingly, the protein production levels diverged from gene expression levels, with the VSCM group consistently presenting a higher value of protein detection than the GPP group at all time points. We have attempted to evaluate the absorption and desorption of VEGF by both materials and got quite intriguing results. First, the amount of VEGF in a solution incubated for 6 h with both materials was higher than it was incubated without materials. This finding suggests that both GPP and VSCM seem to protect VEGF from degradation, and VSCM is superior in this ability. Furthermore, the amount of VEGF subsequently released within the next 24 h was significantly higher for VSCM than for GPP. The nature of this difference is unclear and can be explained by the lower ability of GPP to absorb VEGF, as well as by better VEGF release from VSCM. The difference in the releasing process between materials can be due to the strong electrostatic adsorption properties of the GPP electrospun materials [42].

The assorted levels of foreign body reactions or inflammatory responses triggered by external grafting materials are major constraints on their clinical application. IL-8 is a pro-inflammatory chemokine that plays a paramount role in recruiting neutrophils [43]. We found that both scaffolds positively modulated IL-8 gene expression and protein production, although this effect was not always statistically significant. It should be noted that a certain degree of inflammation is an essential part of the initial phase of regeneration and is essential in controlling infections.

The major limitation of this study is in vitro design. Although both types of scaffolds showed promising potential for periodontal regeneration and clinical application, in vitro conditions cannot fully mimic the physiological environment of the inner periodontium structure. Consequently, further in vivo studies are required to verify these findings. Furthermore, the mechanisms by which fibrous materials regulate cellular behavior remain at a relatively rudimentary level. To explore deeper mechanisms, such as whether the discrepancy between results from different materials is due to integrin-mediated cascades or deformation of cell morphology, further specialized studies, which would offer more insight into the future optimization and modification of fibrous structures, are necessary.

## 4. Materials and Methods

### 4.1. Ethics

All experimental protocols strictly followed the ethical guidelines of the Declaration of Helsinki regarding medical research involving human subjects. The procedure for isolating and working with hPDL-MSCs received approval from the Ethics Committee of the Medical University of Vienna, Vienna, Austria (vote no.1079/2019, extended in 2024). Patients gave their written informed consent before tooth donations.

### 4.2. Cell Isolation

Primary hPDL-MSCs were harvested from periodontally healthy third molars, free of inflammation and caries, of five patients undergoing extraction procedure for orthodontic purposes, following a previously described method [44]. Periodontal ligament tissue segments were scrapped from the middle third of the root surface and placed in Dulbecco’s Modified Eagles Medium (DMEM, Sigma-Aldrich, St. Louis, MO, USA), enriched with 10% fetal bovine serum (FBS, Gibco, Carlsbad, CA, USA), 100 U/mL penicillin and 50 μg/mL streptomycin (P/S, Gibco, Carlsbad, CA, USA). Cells outgrew from tissue segments were maintained at 37 °C in an environment with 5% CO_2_ and 95% humidity. Once cells reached confluence, they were detached using Accutase^®^ (Sigma-Aldrich, St. Louis, MO, USA) and expanded in the culture flask. Only cells from the fourth to sixth passages were utilized in the experiments. The phenotype of isolated hPDL-MSCs as mesenchymal stromal cells was verified in accordance with the prior studies [44,45]. In line with the minimal criteria depicted by the International Society for Cell and Gene Therapy (ISCT), more than 97% of cells were positively stained with mesenchymal surface markers, including CD29, CD73, and CD90 and less than 0.5% of cells were positively stained with hematopoietic surface markers CD31 and CD34 [46]. Research manuscripts reporting large data sets that are deposited in a publicly available database should specify where the data have been deposited and provide the relevant accession numbers. If the accession numbers have not yet been obtained at the time of submission, please state that they will be provided during review. They must be provided prior to publication.

### 4.3. Preparation of GPP and VSCM

GPP specimens (Neo Modulus [Suzhou] Medical, Suzhou, China) with 6 mm were prepared using a corneal trephine (Shimei Medical, Shenzhen, China) under sterile conditions. Meanwhile, VSCM (Fibro-Gide^®^, Geistlich Pharma AG, Wolhusen, Switzerland) was sliced into 3 mm thicknesses before 6 mm-diameter specimens were punched out. These specimens were then positioned at the bottom of 96-well plates for experimentation.

### 4.4. Analysis of Cell Morphology and Attachment

The morphology of cells and the microstructure of biomaterials were observed using scanning electron microscopy (SEM) as previously described [24]. hPDL-MSCs were seeded onto GPP and VSCM at the density of 5 × 10^3^/well in 200 μL of DMEM and cultured at 37 °C. After 3, 7, and 14 days, the samples were fixed overnight with 4% paraformaldehyde (ThermoFisher Scientific, Waltham, MO, USA) and washed with PBS to eliminate detached cells. The samples were then dehydrated by gradient ethanol washing. After replacing the remaining ethanol with hexamethyldisilazane (HMDS, Sigma-Aldrich, St. Louis, MO, USA), the sample surfaces were sputter-coated and examined under SEM (FEI Quanta 200, Hillsboro, OR, USA) at an accelerating voltage of 15–20 kV.

The adhered hPDL-MSCs were also visualized by staining the nuclei and cytoskeleton structures using the focal adhesion staining kit (Sigma-Aldrich, St. Louis, MO, USA) according to the manufacturer’s protocol. hPDL-MSCs with an initial seeding density of 5 × 10^3^/well were cultured on GPP, VSCM, and tissue culture plate (TCP) in 200 μL of DMEM for 3, 7, and 14 days. Cells were fixed with 4% paraformaldehyde for 15 min and permeabilized with 0.1% Triton-X100 (Sigma-Aldrich, St. Louis, MO, USA) for 5 min at room temperature. After blocking with 1% bovine serum albumin for 30 min, cells were stained with rhodamine-conjugated phalloidin and 4′, 6-diamidino-2-phenylindole (DAPI) to visualize actin filaments and nuclei, respectively. Staining of focal adhesion was not performed because of the interference of material with the antibody. Cell morphology and attachment were qualitatively analyzed and observed with an ECHO Revolve fluorescence microscope (Echo, San Diego, CA, USA). No quantitative analysis was performed because the images were taken in 2D focal plains for 3D material. Therefore, such quantification will not be appropriate for assessing cell attachment.

### 4.5. Cell Proliferation/Viability and Metabolic ActivityAssay

Approximately 5 × 10^3^ hPDL-MSCs in 200 μL of DMEM were seeded onto GPP, VSCM, and TCP surfaces. Cell proliferation/viability was evaluated with a cell-counting kit (CCK-8, Dojindo Laboratories, Kumamoto, Japan) on days 3, 7, and 14, similar to the previously described method [24]. A total of 20 μL of CCK-8 reagent was added and left to incubate for 4 h. The formazan produced from tetrazolium salt was quantified photometrically at a 450 nm wavelength using a microplate reader (Synergy HTX; BioTek, CA, USA). Metabolic activity was assessed by applying a resazurin-based assay (Sigma-Aldrich, St. Louis, MO, USA). To each culture well containing 200 μL of cell solution, 20 μL of resazurin solution was added and incubated for 6 h. The fluorescence-based plate reader (Synergy HTX; BioTek) was utilized to detect the conversion amount of resazurin to resorufin, with settings at 540/34 nm for excitation and 600/40 nm emission. The tests were carried out through hPDL-MSCs from five individual donors, with each having two technical replicates.

### 4.6. Investigation of Gene and Protein Production

hPDL-MSCs were seeded at a density of 3 × 10^4^/well in 200 μL of DMEM onto three different types of materials. Following incubation periods of 3, 7, and 14 days, the cells were lysed, and their mRNA was transcribed to cDNA utilizing the subsequent TaqMan gene expression assays (Applied Biosystems, Foster City, CA, USA): Collagen type I, alpha1 (COL1A1, Hs00164004_m1), Vascular endothelial growth factor A (VEGF-A, Hs00900055_m1), Periostin (POSTN, Hs01566750_m1), Cementum attachment protein (CAP, Hs00171965_m1), Cementum protein 1 (CEMP1, Hs04185363_s1), Osteocalcin (OCN, Hs01587814-g1), Interleukin 8 (IL-8, Hs00174103_m1), and glyceraldehyde-3-phosphate dehydrogenase (GAPDH, Hs99999905_m1). The reverse transcription reaction was conducted at 37 °C for 1 h, followed by 95 °C for 5 min utilizing the Primus 96 advanced thermocycler (Peq/Lab/VWR, Darmstadt, Germany). Gene expression was quantitatively analyzed with a QuantStudio 3 device (Applied Biosystems, Foster City, CA, USA). All the samples were initially heated at 95 °C for 10 min, then subjected to 50 cycles of denaturation at 95 °C for 15 s and annealing/extension at 60 °C for 1 min. The target gene expression was calculated using the 2^−ΔΔCt^ method, with GAPDH serving as the housekeeping gene and the 3-day TCP group results as the control.

After incubating for 3, 7, and 14 days, VEGF-A and IL-8 protein levels in the conditioned media were measured using a human VEGF-A kit (Cat. Nr. BMS277-2, ThermoFisher Scientific, Waltham, MA, USA) and a human IL-8 ELISA kit (Cat. Nr. 88-8086, ThermoFisher Scientific, Waltham, MA, USA), respectively, following the manufacturer’s instructions and applying the Synergy HTX multi-mode reader (BioTek Instruments, Winooski, VT, USA).

### 4.7. Absorption and Desorption of VEGF by Scaffolds

The scaffolds were incubated in DMEM supplemented with 100 U/mL penicillin and 50 µg/mL streptomycin and containing 10 ng/mL recombinant VEGF protein (Cat. Nr. 100-20-10UG, PeproTech, Cranbury, NJ, USA) for 6 h at 37 °C in an environment with 5% CO_2_ and 95% humidity, then conditioned media were collected. The scaffolds were further incubated for an additional 24 h in DMEM, and conditioned media were collected. The VEGF protein levels in the conditioned medium from both time points were measured using the human VEGF-A kit (Cat. Nr. BMS277-2, ThermoFisher Scientific, Waltham, MA, USA) according to the manufacturer’s protocol. The measurements and data were analyzed using a microplate reader (BioTek Instruments, Winooski, VT, USA) and specialized software (All-In-One microplate reader software Gen5, BioTek Instruments, Winooski, VT, USA). The absorption of VEGF by scaffolds was evaluated based on the difference in the VEGF content in media incubated without and with scaffolds measured at the first time point. The desorption of VEGF was evaluated based on the VEGF content measured in the conditioned media at the second time point.

### 4.8. Statistical Analysis

All quantitative data was summarized as mean ± standard deviation (SD) from a minimum of five independent repetitions. The hPDL-MSCs were obtained from at least five different individuals. Relevant data was analyzed by means of GraphPad Prism software (version 9, La Jolla, CA, USA), with normal distribution verified by the Kolmogorov–Smirnov test. For data following a parametric distribution, one-way analysis of variance (ANOVA) for repeated measures or unpaired *t*-test was used to evaluate differences between groups. When data were not normally distributed, the nonparametric Friedman test was utilized for analysis. Differences were deemed statistically significant at *p*-values < 0.05.

## 5. Conclusions

Summarizing, both materials in the study offered suitable conditions for hPDL-MSCs adhesion. However, resulting from the discrepancy in fiber structure, the attachment and distribution of cells are different. Furthermore, the differences in the expression of various proteins differed in some cases between materials, suggesting that they may promote different aspects of periodontal regeneration. We found that the expression of markers, indicating a potential differentiation into various tissue-specific cells, like periodontal ligament cells, osteoblasts, or cementoblasts, depends on time and material. A possible explanation for this cold is the different scaffold 3D architecture or material composition. In particular, GPP could be superior for the regeneration of the cementum and periodontal ligament compartment, whereas VSCM seems better at supporting bone compartment regeneration. Further studies should evaluate whether these differences occur in in vivo conditions. The construction of a new scaffold combining the properties of both GPP and VSCM could be a feasible approach for future research.

## Figures and Tables

**Figure 1 ijms-26-00672-f001:**
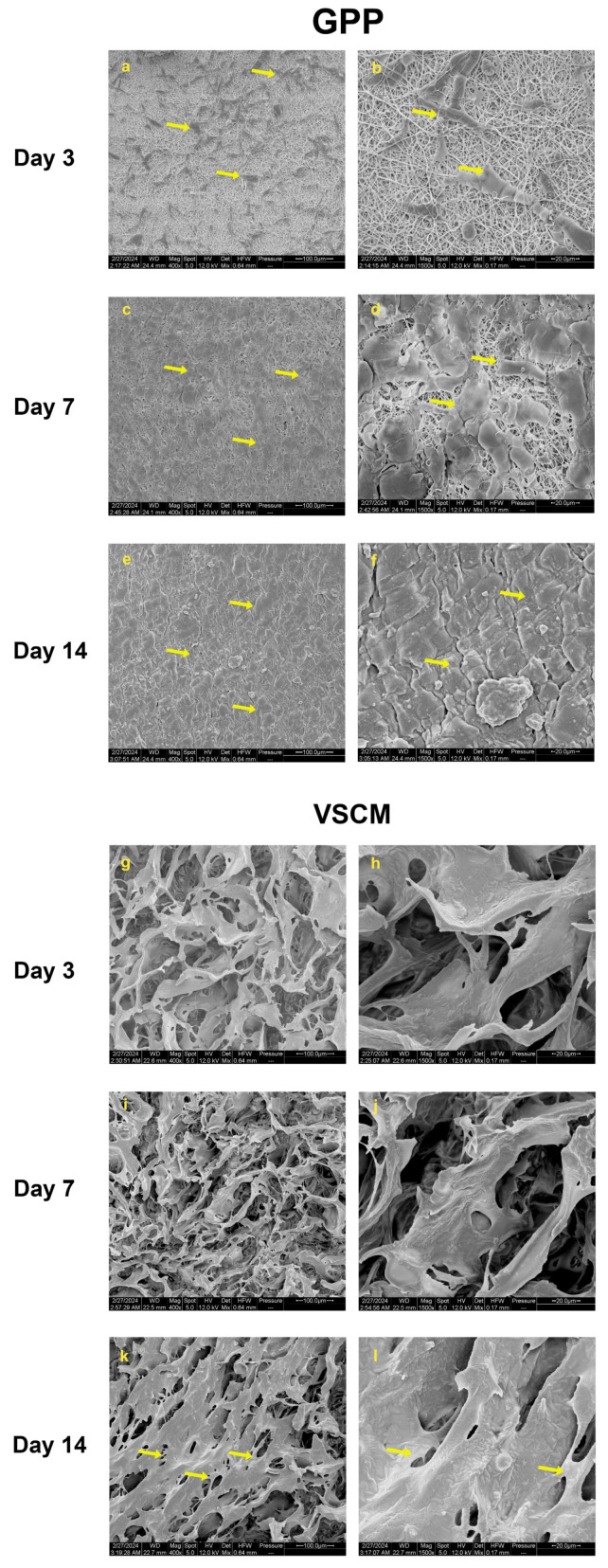
Colonization and patterns of cells observation on two scaffolds through SEM. hPDL-MSCs were grown on GPP (**a**–**f**) or VSCM (**g**–**l**) for 3, 7, or 14 days. The images were taken at magnification 400-fold (**left panels**) or 1500-fold (**right panels**). Scale bars correspond to 100 µm and 20 µm, respectively. The yellow arrows indicate cells. GPP demonstrated nanoscale, microscale fiber, and VSCM a porous structure; cell attachment and growth were well-established on both substrates.

**Figure 2 ijms-26-00672-f002:**
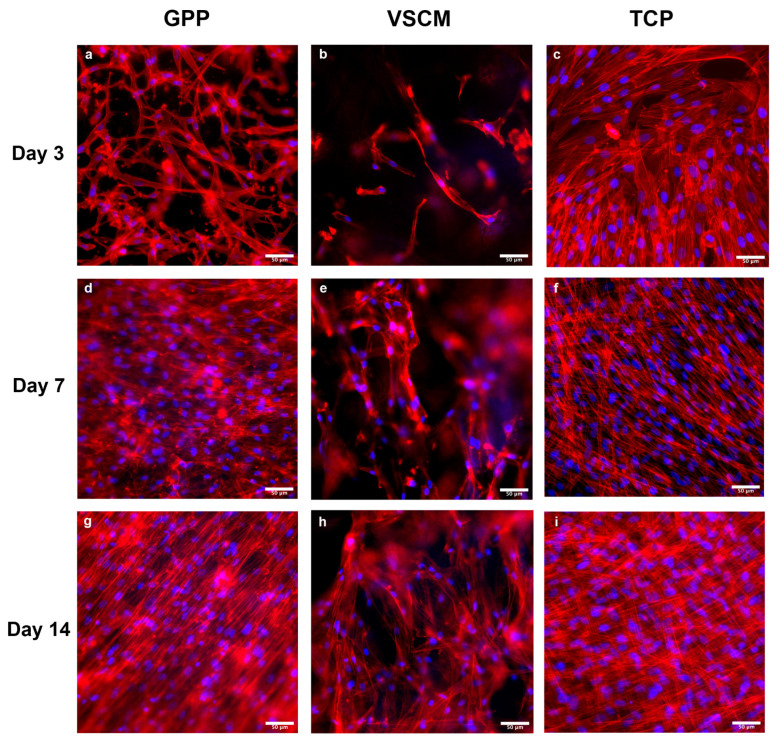
The morphological characteristics of hPDL-MSCs observed on different surface types. hPDL-MSCs were grown on GPP (**a**,**d**,**g**), VSCM (**b**,**e**,**h**) or TCP (**c**,**f**,**i**) for 3, 7, or 14 days and stained with a focal adhesion staining kit. F-actin was stained with TRITC-conjugated phalloidin (red) and the nucleus with DAPI (blue).The images were captured at 100-fold magnification; scale bars correspond to 50 µm. The cells exhibited different morphologies and distributions on the three different materials.

**Figure 3 ijms-26-00672-f003:**
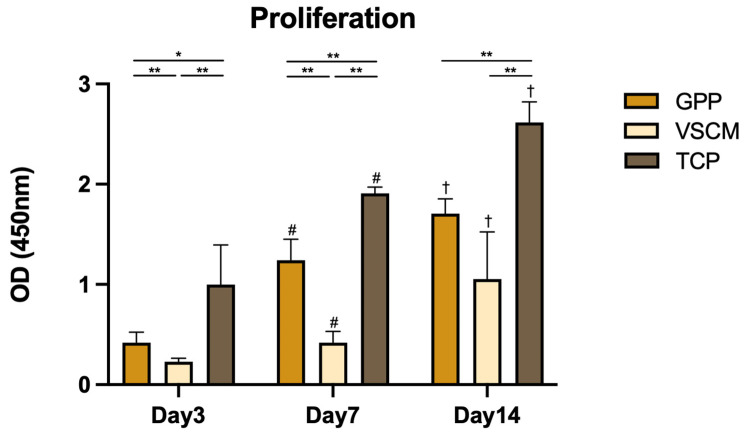
Assessment of the proliferation/viability of hPDL-MSCs when cultured on GPP, VSCM, and TCP surfaces. Cells were cultured on different substrates for 3, 7, and 14 days, and their proliferation/viability was measured using a CCK-8 assay. The Y-axis shows the OD values measured at 450 nm. Data are presented as mean ± SD of five independent experiments with hPDL-MSCs isolated from five different donors. * and **—significantly different between substrates with *p* < 0.05 and 0.01, respectively. #—significantly different between day 3 and day 7 on the same substrate. †—significantly different between day 7 and day 14 on the same substrate.

**Figure 4 ijms-26-00672-f004:**
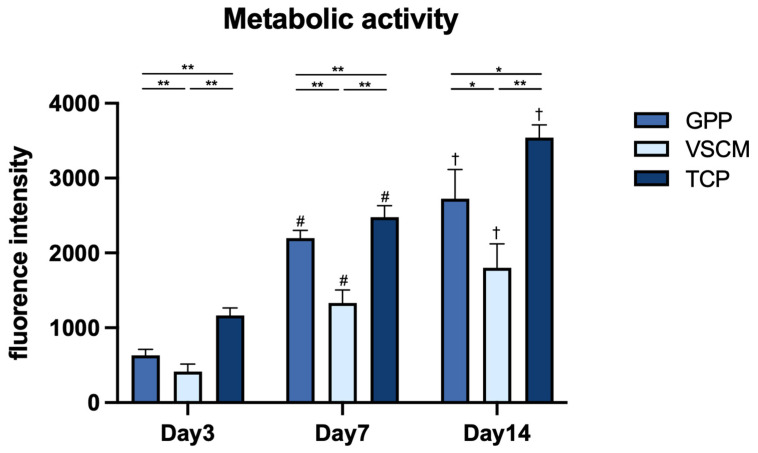
Metabolic activity of hPDL-MSCs grown on GPP, VSCM, and TCP. Cells were cultured on different substrates for 3, 7, and 14 days, and their metabolic activity was measured using a resazurin-based assay. The Y-axis shows the fluorescence measured with the excitation settings at 540/34 nm and emission at 600/40 nm. Data are presented as mean ± SD of five independent experiments with hPDL-MSCs isolated from five different donors. * and **—significantly different between substrates with *p* < 0.05 and 0.01, respectively. #—significantly different between day 3 and day 7 on the same substrate. †—significantly different between day 7 and day 14 on the same substrate.

**Figure 5 ijms-26-00672-f005:**
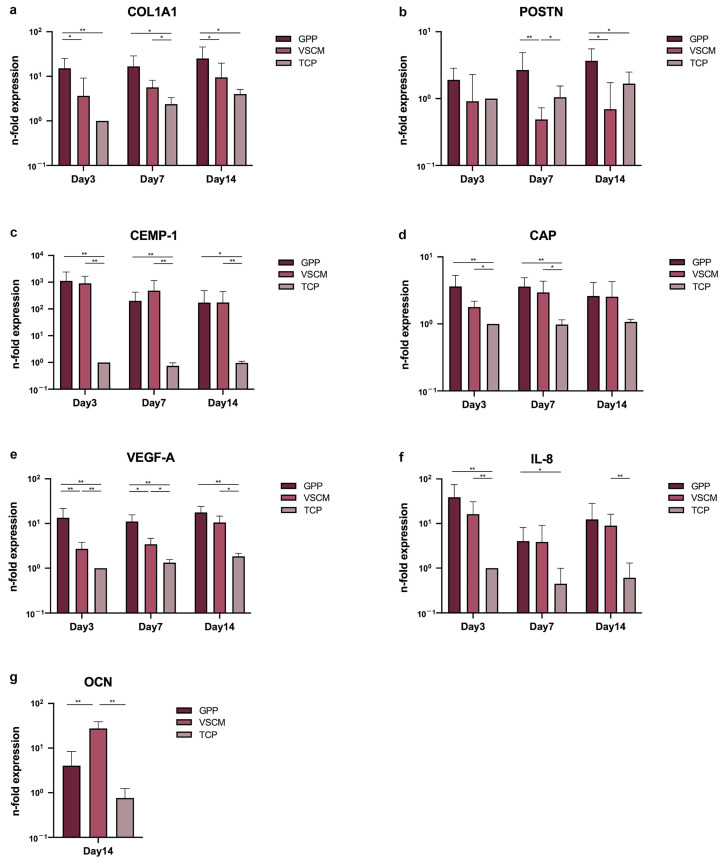
Analysis of gene expression for diverse biomarkers in hPDL-MSCs cultured on different types of materials. hPDL-MSCs were cultured on different substrates for 3, 7, and 14 days, and the gene expression of COL1A1 (**a**), POSTN (**b**), CEMP-1 (**c**), CAP (**d**), VEGF-A (**e**), IL-8 (**f**), and OCN (**g**) was measured by qPCR. Y-axes show *n*-fold expression of the corresponding gene in relation to that measured on tissue culture plastic on day 3 and calculated by a 2^−ΔΔCt^ method using GAPDH as a housekeeping gene. Comparison results of OCN was ultimately limited to day 14, as its values were too low for reliable detection at early time points. Data are presented as mean ± SD of five independent experiments with hPDL-MSCs isolated from five different donors. * and **—significantly different between substrates with *p* < 0.05 and 0.01, respectively.

**Figure 6 ijms-26-00672-f006:**
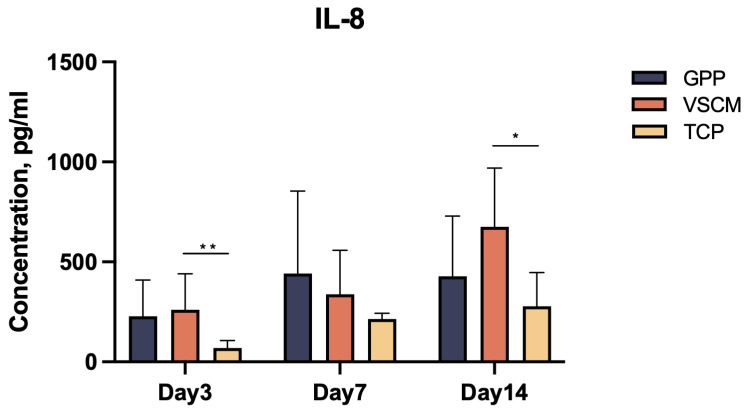
IL-8 protein production by hPDL-MSCs grown on three types of substrates. The content of IL-8 in the conditioned media was measured by ELISA after 3, 7, and 14 days of culture. Data are presented as mean ± SD of five independent experiments with hPDL-MSCs isolated from five different donors. * and **—significantly different between substrates with *p* < 0.05 and 0.01, respectively.

**Figure 7 ijms-26-00672-f007:**
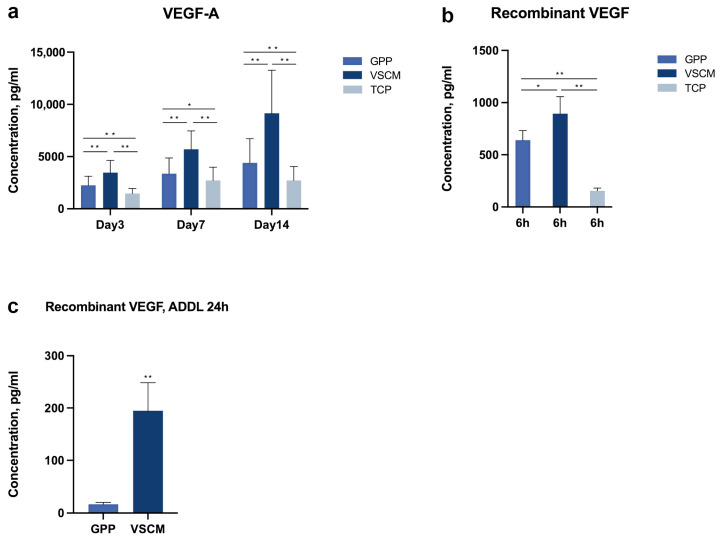
VEGF protein production by cells and absorption by materials. (**a**)—hPDL-MSCs were cultured on different substrates for 3, 7, and 14 days, and the content of VEGF in conditioned media was measured by ELISA. (**b**)—the scaffolds were incubated in a medium containing 10 ng/mL recombinant VEGF protein for 6 h at 37 °C, and the content of VEGF in conditioned media was determined by ELISA at the end of the incubation. (**c**)—after incubation with recombinant VEGF, the scaffolds were incubated for an additional 24 h in a medium, and the amount of released VEGF was measured by ELISA. Data are presented as mean ± SD of five independent experiments with hPDL-MSCs isolated from five different donors. * and **—significantly different between substrates with *p* < 0.05 and 0.01, respectively.

## Data Availability

The data presented in this study are available on request from the corresponding author.

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
