# Peer review of "Potential of Trilayered Gelatin/Polycaprolactone Nanofibers for Periodontal Regeneration: An In Vitro Study"

_ijms, 2025, doi:10.3390/ijms26020672_

Round 1
Reviewer 1 Report
Comments and Suggestions for Authors
Please read the pdf file attached.

Author Response
Reply to the comments of Reviewer 1
We are thankful to this Reviewer for the critical comments on our manuscript. We considered all points and changed the manuscript accordingly. The revised text is highlighted in yellow color. Here are our point-by-point answers:
COMMENT 1
At what time point the differentiation begins, according to the presented results?
AUTHORS’ RESPONSE
In our experiments, we did not investigate a specific differentiation of hPDL- MSCs and would like rather to focus on the expression of various functional proteins. The reason for this is that such an approach better reflects the situation in vivo. For example, osteogenic differentiation in vitro requires an artificial medium with components that are not present in vivo. Therefore, we decided not to focus on a specific differentiation. Instead, we measured the expression of specific markers of cementoblasts, periodontal ligament cells and osteoblasts, but it would be rather speculative to conclude about differentiation based only on these markers. We added some remarks about the association between marker expression and differentiation to the revised version of the manuscript (see P. 11, lines 329-331 and P. 12, lines 341-344; 355-357).
COMMENT 2
When the differentiation begins the metabolic activity presents a reduction. So, the authors supports that the differentiation begins after day 14. Is that correct?
AUTHORS’ RESPONSE
The expression of osteogenic differentiation marker, osteocalcin, was detected only after 14 days of culture. Therefore, we may assume that at this time point, the differentiation of hPDL-MSCs to bone cells might be promoted. However, your suggestion remains crucial, and we have accordingly revised the relevant sections to emphasize this point (see P. 12; lines 355-357).
COMMENT 3
Please try to explain the sequence of the molecular activities and cells reactions
AUTHORS’ RESPONSE
Thank you for this comment. We added this information to the revised manuscript (see P. 11, lines 289-293)
COMMENT 4
Please provide more information concerning the primary cell line [percent of gingival fibroblasts, periodontal ligament cells etch.]
AUTHORS’ RESPONSE
We are routinely doing cell phenotyping, and more than 97 % of cells exhibited the mesenchymal character; this information (referenced to our recent study) is provided in the manuscript (see P. 13, lines 422-424).
COMMENT 5
Could the authors support the construction of a new scaffold – combination of the scaffolds tested- in order to promote periodontal ligament and then bone regeneration?
AUTHORS’ RESPONSE
Thank you for this comment; we added such perspective to the conclusion section (see P. 15, lines 526-528).
COMMENT 6
The main goal is periodontal ligament regeneration or bone regeneration?
AUTHORS’ RESPONSE
Thank you for your insightful questions concerning this content. We evaluated the potential of 2 materials to stimulate periodontal regeneration, which is quite complex mainly because of the heterogeneity of periodontium. Periodontal functional tissues refer to a periodontal complex composed of cementum, periodontal ligament, and alveolar bone, and current research is forwarded to achieve simultaneously regenerate these components into their naturally structured compartments. In our test, GPP exhibited stronger regeneration ability for the periodontal ligament and cementum compartments, whereas its capacity to regenerate the alveolar bone compartment was less effective than VSCM, which was notably weaker in regeneration of the first two compartments, especially the ligament tissue. Thus, both materials showed specific periodontal regenerative capabilities, although their advantages differ. Based on your comment, we added a remark to the conclusion section (see P. 15, lines 523-525).
COMMENT 7
Did the authors investigate scaffolds absorption?
AUTHORS’ RESPONSE
Thank you for this comment. We investigated the absorption and release of VEGF by both materials. These data are presented in Figure 7.
COMMENT 8
Please add arrows at SEM microphotographs to show cells. The presented photos are rather confusing and are not clear where the cells are located.
AUTHORS’ RESPONSE
Thank you for your suggestion. We revised SEM and added arrows to it.
Reviewer 2 Report
Comments and Suggestions for Authors
This article examines the in vitro behavior of periodontal ligament-derived stromal cells used in periodontal tissue engineering (the metabolic effect of the GPP prototype on human periodontal ligament mesenchymal stromal cells).
The topic is original and relevant in the field since periodontal loss is frequent, and surgical treatment requires biocompatible regenerative materials. The study provides a clear presentation of the cell growth process for a period of two weeks following some specific parameters all under fluorescence microscopy.
Introduction section – provides a clear presentation of the problem.
Results section—provide sufficient data to understand the experiment as well as fluorescence microscopy images.
Discussion – the arguments both pro and contra regarding the growth and uses as well as the limits of the study and the need for in vivo study.
Materials section- is comprehensive, but in our opinion should follow the introduction section.
Conclusions -supports the case report presentation.
References- are proper and appropriate
I recommend acceptance as it is.
Author Response
We are thankful to this reviewer for the evaluation of our manuscript and recommendation of acceptance.
Reviewer 3 Report
Comments and Suggestions for Authors
- Figures must write the letter of the different images.
- Figures´ legends must describe properly the results.
- IL-8 should be included in another section different from the VEGF-A.
- Describe the focal adhesion staining kit.
- How were the cells morphology and attachment with the fluorescence microscopy quantified.
Author Response
Reply to the comments of Reviewer 3
We are thankful to this Reviewer for the critical comments on our manuscript. We considered all points and changed the manuscript accordingly. The revised text is highlighted in yellow color. Here are our point-by-point answers:
COMMENT 1
Figures must write the letter of the different images.
AUTHORS’ RESPONSE
Thank you for your comment. We have corrected the letters in all Figures so that they appear uniform throughout the manuscript.
COMMENT 2
Figures´ legends must describe properly the results.
AUTHORS’ RESPONSE
Thank you for this important comment. We have modified the legends of all figures so that they can be understood without referring to the main text.
COMMENT 3
IL-8 should be included in another section different from the VEGF-A.
AUTHORS’ RESPONSE
In the revised version, we separated IL-8 and VEGF-A protein data and split the former Figure 6 into a new Figure 6 and Figure 7A.
COMMENT 4
Describe the focal adhesion staining kit.
AUTHORS’ RESPONSE
The manufacturer of the focal adhesion kit was added to the manuscript (see, p.14, line 447). The details of the staining are also provided in the same paragraph.
COMMENT 5
How were the cells morphology and attachment with the fluorescence microscopy quantified.
AUTHORS’ RESPONSE
We did not perform any quantification; the images were used only to evaluate cell morphology qualitatively.
Round 2
Reviewer 1 Report
Comments and Suggestions for Authors
Dear authors, thank you for the corrections. I have a few more comments.
Please open the attached file.

Author Response
Thank you for the suggestions and questions you provided to improve our manuscript. The revised text is highlighted in yellow color. Here are our point-by-point answers:
COMMENT 1
First, I could not detect any red row showing cell position.
AUTHORS’ RESPONSE
We substituted red with yellow and made the arrows bigger. I hope it is well visible now.
COMMENT 2
Also, I would like to ask to present more information concerning differences in cell morphology (line #148).
AUTHORS’ RESPONSE
Thank you for your suggestions, we have revised the description of cell morphology accordingly (see P. 5; lines 137-142).
COMMENT 3
Finally, do the authors indicate that differentiation begins at day 7 and 14 at the different subgroups? And if this is correct, please provide a possible explanation.
AUTHORS’ RESPONSE
Thank you for your question and suggestion. We added the remark regarding differentiation and possible influencing factors to the conclusion section (see, p. 16, lines 534-537).
Reviewer 3 Report
Comments and Suggestions for Authors
This requirement should be attended:
- How were the cells morphology and attachment with the fluorescence microscopy quantified.
Author Response
Thank you for the suggestions and questions you provided to improve our manuscript. The revised text is highlighted in yellow color. Here are our point-by-point answers:
COMMENT 1
How were the cells morphology and attachment with the fluorescence microscopy quantified.
AUTHORS’ RESPONSE
Thank you for your suggestions. In our current study, the primary purpose of using fluorescence microscopy to observe cells was to assess the difference in cell morphology and distribution on three substrates descriptively. The results obtained showed that the morphology and distribution of cells were different between them, which may also be a key factor affecting cellular behaviors regarding gene expression and protein production. For cell quantification, we investigated cell metabolic activities (CCK-8 and resazurin-based assays).
No quantitative analysis was possible because the images were taken in 2D focal plains for 3D material. Therefore, such quantification will not be appropriate for assessing cell attachment. We also added this information to the Material and Methods section of the revised manuscript (see, p. 15, lines 463-467).
We also added some more detailed descriptions of cell morphology, as requested by another Reviewer (see p. 5; lines 137-142). We hope this helps to resolve this question regarding cell morphology.